# The Efficiency of Neurospheres Derived from Human Wharton’s Jelly Mesenchymal Stem Cells for Spinal Cord Injury Regeneration in Rats

**DOI:** 10.3390/ijms24043846

**Published:** 2023-02-14

**Authors:** Sirilak Somredngan, Kasem Theerakittayakorn, Hong Thi Nguyen, Apichart Ngernsoungnern, Piyada Ngernsoungnern, Pishyaporn Sritangos, Mariena Ketudat-Cairns, Sumeth Imsoonthornruksa, Nattawut Keeratibharat, Rangsirat Wongsan, Ruttachuk Rungsiwiwut, Rangsun Parnpai

**Affiliations:** 1Embryo Technology and Stem Cell Research Center, School of Biotechnology, Institute of Agricultural Technology, Suranaree University of Technology, Nakhon Ratchasima 30000, Thailand; 2Laboratory of Embryo Technology, Institute of Biotechnology, Vietnam Academy of Science and Technology, Hanoi 100000, Vietnam; 3School of Preclinical Sciences, Institute of Science, Suranaree University of Technology, Nakhon Ratchasima 30000, Thailand; 4School of Biotechnology, Institute of Agricultural Technology, Suranaree University of Technology, Nakhon Ratchasima 30000, Thailand; 5School of Surgery, Institute of Medicine, Suranaree University of Technology, Nakhon Ratchasima 30000, Thailand; 6The Center for Scientific and Technological Equipment, Suranaree University of Technology, Nakhon Ratchasima 30000, Thailand; 7Department of Anatomy, Faculty of Medicine, Srinakharinwirot University, Bangkok 10110, Thailand

**Keywords:** human umbilical cord Wharton’s jelly-derived mesenchymal stem cells, neurospheres, transplantation, recovery, spinal cord injury

## Abstract

Spinal cord injury (SCI) causes inflammation and neuronal degeneration, resulting in functional movement loss. Since the availability of SCI treatments is still limited, stem cell therapy is an alternative clinical treatment for SCI and neurodegenerative disorders. Human umbilical cord Wharton’s jelly-derived mesenchymal stem cells (hWJ-MSCs) are an excellent option for cell therapy. This study aimed to induce hWJ-MSCs into neural stem/progenitor cells in sphere formation (neurospheres) by using neurogenesis-enhancing small molecules (P7C3 and Isx9) and transplant to recover an SCI in a rat model. Inducted neurospheres were characterized by immunocytochemistry (ICC) and gene expression analysis. The best condition group was selected for transplantation. The results showed that the neurospheres induced by 10 µM Isx9 for 7 days produced neural stem/progenitor cell markers such as Nestin and β-tubulin 3 through the Wnt3A signaling pathway regulation markers (*β-catenin* and *NeuroD1* gene expression). The neurospheres from the 7-day Isx9 group were selected to be transplanted into 9-day-old SCI rats. Eight weeks after transplantation, rats transplanted with the neurospheres could move normally, as shown by behavioral tests. MSCs and neurosphere cells were detected in the injured spinal cord tissue and produced neurotransmitter activity. Neurosphere-transplanted rats showed the lowest cavity size of the SCI tissue resulting from the injury recovery mechanism. In conclusion, hWJ-MSCs could differentiate into neurospheres using 10 µM Isx9 media through the Wnt3A signaling pathway. The locomotion and tissue recovery of the SCI rats with neurosphere transplantation were better than those without transplantation.

## 1. Introduction

Every year, between 250,000 and 500,000 people around the world (World Health Organization; WHO) suffer from spinal cord injuries (SCI). Generally, spinal cords are injured by mechanical factors, including falls, motor vehicle accidents, sports injuries, work-related injuries, and community violence. After SCI, pathological mechanisms start, followed by inflammation and cell degeneration in the injured region [1]. SCI pathology can be divided into two phases. The primary phase begins with a direct traumatic effect on the neural cells and vascular structures of the spinal cord. The secondary phase involves inflammatory activation leading to cellular apoptosis, necrosis, glial scar formation, and prolonged Wallerian degeneration. This degeneration process happens whenever a nerve fiber is cut or crushed at the axon distal to the injured part [2]. SCI also causes several other severe medical issues, such as respiratory and urogenital issues, and skin abnormalities [3]. Furthermore, young SCI patients often require prolonged medical and social care leading to significant socioeconomic issues [4]. Normally, steroids are used with SCI patients to limit existing cell death. To date, stem cell therapy has played an essential role in the clinical treatment of neurodegenerative disorders because stem cells have high proliferative potential, self-renewal, and the ability to differentiate into various types of cells [5].

Mesenchymal Stem Cells (MSCs) are multipotent stem cells that have the potential to multiply on their own and the ability to differentiate into many different types of cells; for example, bone cells, cartilage cells, liver cells, pancreatic cells, fat cells, neuronal cells, and glial cells [6,7]. MSCs also have immunosuppressive properties which are important for the allogeneic cell therapy process [8,9,10]. MSCs release large amounts of brain-derived neurotrophic factor (BDNF), neurotrophin-3 (NT-3), and nerve growth factor (NGF), as well as other neuroprotective compounds, in addition to stimulating neurogenesis and angiogenesis [11]. Significantly, MSCs could induce tissue regeneration and prevent the development of tumors associated with stem cell transplantation [12]. MSCs are found in many tissues; for example, umbilical cord, bone marrow, placenta, and adipose tissues [13]. However, depending on the age of the tissue samples, they may provide a small number of cells with low proliferation potential. In addition, a high chance of virus infection can be found [14,15,16]. On the other hand, MSCs derived from human Wharton’s jelly (hWJ-MSCs) are easily obtained and normally discarded. hWJ-MSCs also have high proliferation capability. In the case of transplantation, hWJ-MSCs do not change into carcinogenic or teratogenic cells, and are non-tumorigenic [17,18]. Besides, hWJ-MSCs demonstrated the characteristics of both embryonic stem cells and adult stem cells. Thus, they could be a suitable source for therapeutic use of stem cells [19]. MSCs can also be transdifferentiated into neurospheres (NSs). NSs are spheres formed of neural stem/progenitor cell aggregates with the ability to differentiate into neuronal lineages [20,21]. In previous literature, NSs from fetal spinal cord cultures have been transplanted and treated for SCI in rats [22]. Furthermore, SCI rats transplanted by the induction of neural stem/progenitor cells showed improved functional development following SCI recovery [23].

Many small molecules have been used to induce human skin fibroblasts to become neurons. P7C3-A20 and Isx9 demonstrated high efficiency for neuron induction [24]. P7C3-A20 has also been shown to activate phosphatidylinositol 3-kinase (PI3K)/protein kinase B (AKT)/glycogen synthase kinase (GSK) in β-catenin signaling [25]. The PI3K/AKT pathway also induces cell proliferation, differentiation, and apoptosis, while GSK—serine/threonine-protein kinase—has been shown to induce apoptosis prevention in brain injury models [26]. Isoxazole 9 (Isx9; N-cyclopropyl-5-(thiophene-2-yl) isoxazole-3-carboxamide) is another small molecule that strongly induced adult neural stem cells to neuronal linage in vitro [27]. Furthermore, Isx9 stimulates the myocyte-enhancer factor 2, a family of transcription factors [28], which impacts embryo and early post-natal periods as pro-neuronal and pro-survival factors [29]. In vivo, Isx9 also induced hippocampal neurogenesis, development, and memory by regulating the cell-intrinsic molecular pathway in adult neurogenesis [30].

In this study, hWJ-MSCs were differentiated into NSs using P7C3-20 and Isx9 small molecules to regulate and enhance neurogenesis. The best NSs induction group based on marker expression and timing was selected for transplant to recover SCI in a rat model.

## 2. Results

### 2.1. MSCs Characterization

Characteristics of MSCs, including morphology, surface marker expression, and differentiation potential, were determined. After hWJ-MSCs isolation for 10–14 days, cells were expanded and passaged. At passage 5, cell morphology of two cell lines was similar to fibroblast monolayer cells (Figure 1(Ab,g)). Cell surface marker expressions of MSCs were analyzed by flow cytometry (Figure 1B). Cell line 1 showed positive results for CD73 cell surface markers (99.51%), CD90 (99.77%), and CD105 (99.72%) but negative results for CD34 (0.00%) and CD45 (0.01%). Meanwhile, in cell line 2, the results were positive for CD73 (99.79%), CD90 (99.60%), and CD105 (2.42%),8776514r/negative for CD34 (0.00%) and CD45 (0.01%).

Differentiation of MSCs into osteocytes, chondrocytes, and adipocytes was carried out by using an induction medium for 21 days (Figure 1(Ac–e,h–j)). Only MSCs from cell line 1 could be differentiated to osteocytes, chondrocytes, and adipocytes. Osteocyte differentiation results for cell line 1 showed cells with intracellular calcium accumulation stained with Alizarin Red. The adipocyte differentiation results of cell line 1 showed the intracellular lipid droplets in the cells by Oil Red O staining. For chondrocyte differentiation, both cell lines could differentiate into chondrocytes as tested by the presence of an Alcian Blue-stained region.

Cell line 1 thus passed the MSCs characterization standards from the International Society for Cell & Gene Therapy; therefore, this cell line was used in further experiments.

### 2.2. P7C3 and Isx9 Cytotoxicity

Cell viability of small molecules, P7C3 and Isx9, was evaluated by MTT assay (Figure 2). The results of cell viability were compared with not adding any small molecules. The results found the lowest cell viability of MSCs at 2 µM P7C3 and 5 µM Isx9, was significantly lower than the control (*p* < 0.05). However, the concentrations of 3 µM P7C3 and 10 µM Isx9 were not toxic to MSCs as they were used for neuron induction [24]. Therefore, we selected P7C3 at 3 µM and Isx9 at 10 µM for further induction of MSCs into the NSs.

### 2.3. Neurosphere Characterizations

#### 2.3.1. Immunocytochemical Staining (ICC)

The intracellular antigen-specific neurons, such as Nestin, SOX2, β-tubulin 3, β-catenin, and DCX of the NSs were examined and compared between groups (Figure 3, Figure 4, Figure 5, Figure 6, Figure 7 and Figure 8). Staining and intensity of immunofluorescence (IF) expression results showed that MSCs were induced to be NSs. The results of β-catenin staining (Figure 3 and Figure 8A) revealed that the NSs of all groups had high expression, with no significant difference found in any of the experimental groups. The SOX2 protein for neural stem cell differentiation inhibitor (Figure 4 and Figure 8B) in the P7C3 groups was expressed significantly higher on day 14 than on day 3 (*p* < 0.05). However, the Nestin protein expressions were progressively high in all groups (Figure 5 and Figure 8C) particularly in the Isx9 group, which, on day 14, was much higher than other groups on day 3. Similarly, the results of β-tubulin 3 antibody staining (Figure 6 and Figure 8D) on day 14 of the Isx9 group were higher than for day 3 of the other groups. Finally, DCX staining results (Figure 7 and Figure 8E) for immature neurons showed no significant difference in all experimental groups.

#### 2.3.2. Gene Expression Analysis

Gene expression analysis of NSs by quantitative polymerase chain reaction (qPCR) of *β-catenin*, *NeuroD1*, *SOX2*, *Nestin*, *β-tubulin 3*, and *DCX* genes was compared between each group and the SHSY-5Y neuroblastoma (positive control for neuronal cells). Figure 9A depicts the expression of the *β-catenin* gene. Every group displayed high levels of *β-catenin* gene expression; however, in the P7C3 group on days 7 and 14, the expressions were significantly lower than on day 3 of the Isx9 group (*p* < 0.05), but no significant difference was found between the other groups. The *NeuroD1* gene expression (Figure 9B) in the P7C3 group was significantly lower on days 3 and 7 than on day 10 of the Isx9 group and on day 14 of the control (*p* < 0.05), but not in the other groups. In the meantime, *SOX2* gene expression (Figure 9C) was significantly higher in the control and P7C3 groups on day 14 than in the other groups (*p* < 0.05), but there was no significant difference with the control group on day 7, the Isx9 group on days 10 and 14, the P7C3+Isx9 group on day 3, and the SHSY-5Y group. The expression of the *Nestin* gene for neural stem/progenitor cells is shown in Figure 9D. *Nestin* gene expression was significantly lower on day 3 of all the groups, day 7 of the control group, and days 10 and 14 of the P7C3 group than it was for the Isx9 group on days 10 and 14 and the P7C3+Isx9 group on day 10 (*p* < 0.05). Figure 9E represents the expression of the *β-tubulin 3* gene in neuronal cells. The *β-tubulin 3* gene expression was significantly lower in the control and P7C3 groups on day 14 than in the P7C3 group and the P7C3+Isx9 group on day 3, the SHSY-5Y group, and all days of the Isx9 group (*p* < 0.05), but there was no significant difference in the other groups. The last gene expression results, *DCX* gene expression (Figure 9F), were significantly lower in all groups than the SHSY-5Y group (*p* < 0.05), but there was no significant difference in the other groups.

Nevertheless, the NSs group with the highest levels of *β-catenin*, *NeuroD1*, *Nestin,* and *β-tubulin 3* expression, and with the lowest *SOX2* expression, was chosen for transplantation. According to the results of the gene expression analysis, control on day 10, and the Isx9 and P7C3+Isx9 groups on days 7–14 showed higher expressions of *β-catenin*, *Nestin*, *NeuroD1,* and *β-tubulin 3* genes than the other groups did. Furthermore, those groups also expressed low *SOX2*. This information showed that Isx9 added to the induction medium can affect neurosphere differentiation on day 7. On the other hand, with or without P7C3 in the induction medium does not affect the markers of neural stem/progenitor cells. The early stage of the neurosphere that can be differentiated from hWJ-MSCs for only 7 days will be helpful for stem cell transplantations in a short time. Consequently, we selected the NSs from day 7 of the Isx9 group for further neurosphere transplantation in rats.

### 2.4. Behavioral Tests

Behavioral tests were performed prior to the spinal cord injury induction, before transplantation, and 8 weeks post-transplantation with the Basso, Beattie and Bresnahan (BBB) locomotor rating scale for 5 min [31,32,33]. The results of the average BBB locomotor rating scale are shown in Figure 10. The normal saline group showed a very low score of moving behavior from week 1 to week 7 post-transplantation and a significant difference when compared with the sham and NSs transplanted groups (*p* < 0.05). However, the BBB score results of the normal saline group were significantly lower than the MSCs transplanted group at 1, 2, 3, 5, and 7 weeks post-transplantation (*p* < 0.05). In other words, the group transplanted with NSs demonstrated that spinal cord tissue was restored, allowing them to move similarly to a normal rat (sham group) from the first week post-transplantation.

### 2.5. Histology

#### 2.5.1. Immunofluorescence and Immunohistochemistry Staining

Spinal cord tissues of all groups were examined at 8 weeks after cell transplantation with intracellular antigen-staining specific to neurons. IF was stained with β-tubulin 3, HuNu, MAP2, GFAP, Nestin, ChAT, NF-L, and Olig2 (Figure 11, Figure 12 and Figure 13). Both the MSCs and NSs transplantation groups were found HuNu-stained, indicating that the transplanted cells survived when co-stained with neuronal antigens (β-tubulin 3) in the injured spinal cord tissue (Figure 11A). However, the expression of β-tubulin 3, Nestin, MAP2, NF-L, and Olig2 did not differ between groups. The expression of GFAP for astrocytes was found to be positive in the normal spinal cord (sham group) around the spinal cord tissue. Except for the normal saline group, and the MSCs and NSs transplantation groups, astrocytes were found around the injured cavities of the spinal cord tissue. Moreover, glial scars were observed to have formed around the injured site of the spinal cord tissues (Figure 11B). Expression of ChAT for acetylcholine as a neurotransmitter was found to be higher in the sham, MSCs, and NSs transplantation groups than in the other groups (Figure 12). ChAT results showed that the MSCs and NSs transplantation groups have neurotransmitters similar to the sham group.

In a part of the brain-derived neurotrophic factor (BDNF) for immune responsibility and neuroregeneration, all groups showed greater BDNF expression in white matter (Figure 14A,B). The gray matter displayed high BDNF expression in the control group, before cell transplantation (Figure 14A,C). BDNF expression was still high after 8 weeks of transplantation of MSCs and normal saline groups. Nonetheless, 8 weeks after NSs transplantation, BDNF showed lower expression than the MSCs transplantation group (*p* < 0.05). According to the BDNF results, the immune system response in spinal cord injury rats was reduced 8 weeks after NSs transplantation.

#### 2.5.2. Cavitation Analysis

The results of cavitation analysis from LFB/H&E staining of spinal cord injury tissue are shown in Figure 15. The cavities and shrinkage of the spinal cord showed damage to the neurons from injury. Normal saline, MSCs, and NSs groups showed total areas of the spinal cord that were significantly smaller than the sham group (Figure 15B) (*p* < 0.05). However, the NSs transplantation group showed a small percentage of spinal cord injury space (% area cavitation) and a percentage of the volume of the SCI (% volume cavitation), which showed no significant difference when compared with the sham group, as shown in Figure 15C,D. In contrast to the normal saline group, the percentage of area cavitation and the percentage of volume cavitation were significantly higher than the sham group (*p* < 0.05).

## 3. Discussion

In this study, hWJ-MSCs were isolated and expanded from 2 cell lines. Only cell line 1 passed the MSCs characterization and was use in further experiments. The MSCs line 1 was able to differentiate into NSs by using neurogenesis-enhancing small molecules, P7C3-A20 and Isx9. After differentiation, on day 3, the control and Isx9 groups showed the highest number of NSs (Appendix A). Similarly, high numbers of small NSs formed within 3–5 days from human dental pulp stem cells (hDPSCs) differentiation were observed [25]. The average diameter of primary NSs was 97 µm, and they were capable of self-renewal. To characterize the NSs, gene and protein expression were used in this study.

Immunocytochemistry (ICC) was used to characterize the NSs. The result showed a gradual increase in Nestin expression on day 3 and a peak on day 14 in the Isx9 group. Nestin expression showed the ability of MSCs to differentiate into neural stem/progenitor cells [26], similar to the result of NSs differentiated from hDPSCs at 14 days that also found high expression of Nestin [25]. For SOX2, results showed the highest expression on day 14 in the P7C3 group. However, SOX2 expression was inhibited via the Wnt3A signaling pathway from many signals in the induction medium, such as bFGF, P7C3, and Isx9 [27,28]. Although the results of β-catenin and DCX staining showed the expression in all experimental groups, no NeuroD1 protein was found in any of the groups on day 7 (Appendix A). β-catenin protein is activated by a variety of signaling pathways involved in proliferation and differentiation. Moreover, the Wnt3A signaling pathway followed by NSCs differentiation and Nestin protein expression needs specific signals and timing to progress toward terminal differentiation or maturation [27,29]. Similarly, β-tubulin 3 protein expression in neuronal cells showed a gradual increase, with the highest expression on day 14 in the Isx9 group but no significant difference when compared with other induction durations. β-tubulin 3 protein expression was displayed on day 14 in this study.

Furthermore, all gene expressions of *β-catenin*, *Nestin*, *NeuroD1,* and *β-tubulin 3* genes in the Isx9 group at 7 days were higher than the other groups. Similarly, the previous results showed high expression of *Nestin* as a multiple neurogenic gene after induction. Besides, a previous study reported that the transcriptional pattern of NSs was differentiated from both MSCs and human neural stem cells (hNSCs) [30]. However, *Nestin* might not identify just neural stem/progenitor cells but may also be expressed in neuroectoderm-like cells [34]. As a result, the outcomes of Nestin expression were not the primary focus of this experiment. Previous reports showed that the induction of MSCs into NSs had high *β-tubulin 3* gene expression, beginning on day 4 of induction, that could predict transcriptional regulation of adult neurogenesis [35]. Similarly, the results at 7 days of the Isx9 group also expressed *β-tubulin 3*, *β-catenin,* and *NeuroD1* genes, which is an important transcription factor involved in neuronal development and maturation. The production of the *NeuroD1* genes such as Nestin and β-tubulin 3 protein was also found in NSs from the Isx9 group at 7 days. For neurogenesis, Wnt3A signaling (Figure 16) prevents the SOX2-dependent regulation of *NeuroD1* gene transcription [27,36]. As a result, low SOX2 marker (repressor) expression is an important point of NSs induction by Isx9 at 7 days. From the gene expression results, we concluded that early-stage NSs can be differentiated from hWJ-MSCs after 7 days of induction in the media supplemented with 10 µM Isx9. These NSs can be beneficial for stem cell transplantation in a short time. Furthermore, MSCs cultured in 3D conditions for more than 7 days may increase the number of dead cells inside the spheroid [37]. Thus, we selected the NSs from day 7 of the Isx9 group for further NSs transplantation in rats.

The NSs used for in vivo transplantation treatment contributed to recovery in the clip compression SCI model. The clip compression model is the most closely related to human SCI, which is primarily caused by burst compression fractures and dislocation [38]. Eight weeks after transplantation, the transplanted cells differentiated into neural lineages including neurons and oligodendrocytes, as detected by specific markers including β-tubulin 3 and HuNu. Moreover, Nestin, ChAT, MAP2, GFAP, NF-L, and Olig2 were also evaluated in the SCI tissues. HuNu-stained transplanted cell survival and neuronal antigens (β-tubulin 3) were found in the injured spinal cord tissues of MSCs and NSs transplantation groups. As previously reported [13,39], MSCs transplantation into neurological disease animals also showed recovery of the nervous system. β-tubulin 3 (neuron cells), Nestin (neural stem/progenitor cells), MAP2 (neurons), NF-L (neuron), and Olig2 (mature oligodendrocytes and myelinating Schwann cells) were found to be expressed in every group. Similarly, embryonic stem cell-derived neurospheres (ESC–NS) were transplanted into the acute and sub-acute phases of the SCI model. One to two weeks post-transplantation, specific markers including Nestin, Olig2, MAP2, astrocytes, and oligodendrocytes were also detected in the spinal cord tissue [26]. The expression of ChAT for acetylcholine—a neurotransmitter—was found to be higher in the sham, MSCs, and NSs transplantation groups than in the other groups. The results showed that the MSCs and NSs transplantation groups had neurotransmitters similar to those of the sham group [40]. Furthermore, positive GFAP expression in astrocytes was spread to all the areas in the normal spinal cord (sham group). Unlike the normal saline, MSCs, and NSs transplantation groups, astrocytes were only found around the injured cavities of the spinal cord tissue.

GFAP expression is seen in the acute-subacute stage of SCI from 2 days to 2 weeks after injury which becomes a physical and chemical barrier to axonal regeneration via astrocytic scar or glial scar. Therefore, a glial scar always forms around the injured site of the SCI tissue [41,42,43]. Astrocytes also help to recover the ionic homeostasis and the integrity of the blood-brain barrier, which helps reduce edema and immune cell invasion [44].

The cavities and shrinkage of the spinal cord showed damage to the injured neurons by cavitation analysis. Normal saline, control, MSCs, and NSs transplantation groups showed a smaller total spinal cord area than the sham group did. However, only the normal saline group showed the highest percentage of the area and volume of cavitation. On the other hand, the NSs transplantation group showed the total area of spinal cord, percentage of the area, and the volume of cavitation of the spinal cord close to the sham group and the MSCs group. The results of this study showed that MSCs and NSs transplanted groups reduced SCI space compared with the normal saline group. Preferably, the NSs transplantation group contributed to the recovery of the SCI. However, transplanted cells did not only recover the neuronal network but were also able to reduce inflammation, regenerate axons and synapses, and reduce glial scars, which could be achieved through cell replacement, functional multipotency, and especially stem cell regeneration [45].

Neurotrophic factors, such as BDNF, NT-3, and NGF, have also been implicated in neurorepair/neuroprotection and immune system responses [46]. According to a previous study, NSs transplantation after SCI for 48 h showed immune response and neuroregeneration with high BDNF levels [47]. Similarly, as previously reported [12], transplantation of MSCs from human placental cords co-cultured with neurons from the human spinal cord into a SCI rat model also showed high BDNF expression. Neurotrophic factors may restore functions through cell transplantation after SCI. Neurotrophic factors secreted by stem cells improve cell survival, differentiation, neuroprotection, remyelination, axonal sparing, axonal growth, neuronal survival, and synaptic plasticity with improved motor functional recovery from neuroprotection, which plays an important role in both the PNS and CNS [46]. Furthermore, the brain blood barrier (BBB) induced an inflammatory response as well as an enhancement of astrocytes, T cells, neutrophils, microglia, and monocytes after SCI. High expression of BDNF is produced and expressed by T cells to inhibit the progression of neuronal injury or to promote the persistence of inflammation. Thus, BDNF expressions inhibit T-lymphocyte apoptosis, leading to a chronic inflammatory process [48]. In this study, BDNF showed greater expression in the white matter of SCI tissue in all groups. On the other hand, gray matter displayed high BDNF expression in the control group (before cell transplantation). The BDNF expression was still high after MSCs and normal saline transplantation. Nonetheless, 8 weeks after NSs transplantation, BDNF showed lower expression than the MSCs transplantation (*p* < 0.05). This result related to cavity damage and SCI shrinkage following NSs transplantation. After NSs transplantation, NSs might associated with BDNF level to induce neuroprotection, remyelination, axonal sparing, etc. Then, 8 weeks later, the BDNF down-regulation level was found after improved motor functional recovery as shown in the highest score of BBB locomotor tests. Meanwhile, NSs may be associated with BDNF-level expression in order to inhibit cell rejection after transplantation and delay the chronic inflammatory process [49]

The BBB locomotor tests were performed before and every week after transplantation. The results showed that the average BBB locomotor rating scale of the normal saline group had very low scores of moving behaviors at 1 week until 7 weeks post-transplantation, and statistically significant differences were found when compared with both the sham and NSs transplantation groups. This study revealed improvements in functional outcomes with treatments involving MSCs and NSs transplantation groups and SCI tissue restoration. Likewise, as previously reported, human umbilical cord-MSCs were transplanted into SCI rats. Those rats showed better recovery movement than without cell transplantation [12]. From the first week to 8 weeks after transplantation, the NSs transplanted group demonstrated that spinal cord tissue and hindlimb motor were recovered, similar to the normal rat (sham group). Likewise, olfactory epithelium cells from rats were induced to globose basal cells with NSs formation and then transplanted into SCI rats (contusion model). The result showed hindlimb motor recovery in rats at the end of 8 weeks after transplantation [21]. However, the MSCs transplanted group showed the regeneration of movement at 1 day before transplantation. The possible reason for this might be that a short period (9 days) of SCI induction was used in this study. Similar to previous data from the subacute stages (2 days–2 weeks) SCI found that astrocytes support recovery by restoring ionic homeostasis and the integrity of the BBB, which reduces edema and immune cell invasion [44].

## 4. Materials and Methods

### 4.1. Ethics Statement

The study was conducted in accordance with the Declaration of Helsinki, and the protocol was approved by the Ethics Committee for Research Involving Human Subjects, Suranaree University of Technology (EC-61-58). Human umbilical cord tissues were harvested and preserved aseptically after delivery with informed consent from patients at the Maharat Nakhon Ratchasima Hospital, Nakhon Ratchasima, Thailand.

### 4.2. Reagents

Chemicals used in this research were from Sigma-Aldrich Corporation (St. Louis, MO, USA); antibodies were from Merck Ltd. (Darmstadt, Germany); the cell culture media were from Gibco (Paisley, UK); and plastic cell culture devices were from SPL life sciences (Gyeonggi-do, Republic of Korea).

### 4.3. hWJ-MSCs Isolation and Culture

Two samples of human umbilical cord were isolated. Human umbilical cords around 7–10 cm long were cleaned and both sides were cut off. After that, umbilical cord vessels were separated. The gelatinous tissue surrounding the vessels was collected, chopped into small pieces (3 mm^2^), plated on sterile 90 × 15 mm^2^ culture dishes, and left for 3–5 min at room temperature (RT) to enable the tissue to attach to the dish. The MSCs culture medium (alpha modification of Eagle’s medium (α-MEM) supplemented with 1 mM L-glutamine, 10% fetal bovine serum (FBS), 100 units/mL penicillin, and 100 µg/mL streptomycin) was carefully added to the culture dish. Tissues were cultured in a humidified atmosphere of 5% CO_2_ in air at 37 °C for 10–14 days. The medium was changed every 3 days. MSCs were expanded until passage 3, and either directly used for experiments or cryopreserved with 10% dimethyl sulfoxide (DMSO) and kept in liquid nitrogen.

### 4.4. Flow Cytometric Analysis of hWJ-MSCs

Flow cytometric analysis was performed to confirm the MSCs surface markers. Two cell lines of hWJ-MSCs were performed. The adherent MSCs cells were treated with trypsin-EDTA and collected after being centrifuged at 350× *g* for 5 min. MSCs were incubated in antibodies in the dark for 20 min. All antibodies are listed in Appendix A. Each antibody was diluted in 100 mL of PBS(-)—the following antibodies: anti-CD34-PE (dilution 1:10, Beckman Coulter, Brea, CA, USA), anti-CD45-FITC (dilution 1:20, Biolegend, San Diego, CA, USA), anti-CD73-APC, anti-CD90-APC/A750, and anti-CD105-PE (dilution 1:100, Biolegend). The samples were analyzed by an Attune™ NxT Flow Cytometer (Attune™ NXT, Thermo Fisher Scientific, Cleveland, OH, USA). The percentage of CD34-, CD45-, CD73+, CD90+, and CD105+ positive or negative cell populations were calculated using the FCS Express™ Software.

### 4.5. Trilineage Differentiation Ability of hWJ-MSCs

As described previously [50], MSCs at passage 5 of two cell lines were cultured until 70% confluence in 6-well culture plates coated with 0.1% gelatin. To induce osteogenic lineage, the induction medium was an α-MEM medium supplemented with 100 nM dexamethasone, 0.2 mM L-ascorbate-2-phosphate, 10 mM β-glycerophosphate, 100 units/mL penicillin, and 100 µg/mL streptomycin. The induction medium was replaced every 2 days and cultured for 21 days. Calcium deposits from the cells were stained and visualized by Alizarin Red staining.

To induce adipogenic lineage, MSCs were cultured in the α-MEM medium supplemented with 5% FBS, 10 µg/mL insulin, 100 µM indomethacin, 1 µM dexamethasone, 0.5 mM isobutyl methylxanthine (IBMX), 100 units/mL penicillin, and 100 µg/mL streptomycin. The IBMX was removed from the medium after 1 week of induction. The medium was replaced every 2 days and cultured for 21 days. Cells were stained with Oil Red O to observe oil droplets.

To induce chondrogenic lineage, MSCs were cultured in the α-MEM medium supplemented with 1% FBS, 1% Insulin-Transferrin-Selenium-Ethanolamine (ITS-X), 50 µg/mL ascorbate-2-phosphate, 40 µg/mL L-proline, 100 µg/mL sodium pyruvate, 100 nM dexamethasone, 10 ng/mL of transforming growth factor beta-3 (TGFβ-3), 100 units/mL penicillin, and 100 µg/mL streptomycin. The medium was replaced every 2 days and cultured for 21 days. Glycosaminoglycans (GAG) production was assessed by Alcian Blue 8x staining.

### 4.6. Analysis of P7C3-A20 and Isx9 Cytotoxicity

One thousand MSCs were cultured in 96-well culture plates in MSCs culture medium for 6h to allow cell attachment. The cytotoxicity of neurogenesis-enhancing small molecules, Isx9 and P7C3-A20 (MedChemExpress, NJ, USA), was evaluated by addition into the culture medium at 0, 1, 2.5, 5, 10, and 20 µM for Isx9, and at 0, 0.25, 0.50, 1, 2.5, and 5 µM for P7C3-A20. The cells were cultured at 37 °C for 48 h in a humidified atmosphere of 5% CO_2_. The cytotoxicity of Isx9 and P7C3-A20 on cell viability was analyzed by a 3 (4,5-dimethytthiazol-2-y/) 2,5-diphenyltetrazolium bromide (MTT; InvitrogenTM, Thermo Fisher Scientific, Massachusetts, USA) assay followed the manufacturer’s instructions.

### 4.7. Neurospheres Differentiation

MSCs cells passage 6 were seeded in 60 × 15 mm petri dish (Greiner Bio-One, Kremsmünster, Austria). Approximately 2.4 × 10^6^ cells were resuspended in neurospheres culture medium; NM [35]. NM was composed of DMEM/F-12 medium supplemented with 2% B27, 1% N2, 20 ng/mL basic fibroblast growth factor (bFGF), 20 ng/mL epidermal growth factor (EGF), 2 µg/mL heparin sodium salt, 100 units/mL penicillin, and 100 µg/mL streptomycin combined with or without 3 µM P7C3 and/or 10 µM Isx9. Cells were cultured in a humidified atmosphere at 37 °C and 5% CO_2_ for 14 days. Twenty-four hours after culture and every 3 days the medium was replaced. The experimental groups were divided into 3, 7, 10, and 14 days after induction (Figure 17A): (1) control; cultured with DMEM/F12 (1:1), (2) P7C3; cultured with NM+3 µM P7C3, (3) Isx9; cultured with NM+10 µM Isx9, and (4) P7C3+Isx9; cultured with NM+10 µM Isx9 for 3 days then changed to NM+10 µM Isx9+3 µM P7C3 up to 14 days. The best NSs differentiation group is selected to transplant to recover SCI in the rat model, based on marker expression and timing.

### 4.8. Neurospheres Characterizations

#### 4.8.1. Immunocytochemical Staining

Neurospheres of days 3, 7, 10, and 14 were filtered with nylon net filters (#NY1H02500 (100 µm) and #NY4H04700 (140 µm), Merck Millipore, MA, USA), and only those 100–140 pm in diameter were selected [51]. The cells were analyzed for the neural stem/progenitor cell expression by ICC. To start with, neurospheres were fixed with 100% methanol for 20 min at −20 °C, permeabilized, and blocked with 2% bovine serum albumin (BSA), 5% normal goat serum, 3 mM sodium azide, and 0.2% Triton-X-100. All antibodies are listed in Appendix A. Cells were incubated for 2 h at RT following by incubation with primary antibodies: anti-Nestin antibody (dilution 1:100; neural stem/progenitor cell marker); anti-SOX2 antibody (dilution 1:100, Bioscience, ma, USA; neural stem/progenitor cell marker); anti-β-tubulin 3 antibody (dilution 1:100; neuron cell marker); anti- β-catenin antibody (dilution 1:100; Wnt3A signaling pathway marker); and anti-DCX antibody (dilution 1:100; immature neuron cells marker) overnight at 4 °C. The next day, neurospheres were incubated in secondary antibodies: Alexa fluor^®^ 488 donkey anti-mouse IgG (dilution 1:1000, Invitrogen^TM^, Thermo Fisher Scientific); Alexa fluor^®^ 594 goat anti-rat 1gG (dilution 1:250, Invitrogen^TM^, Thermo Fisher Scientific); and Alexa fluor^®^ 594 goat anti-rabbit 1gG (dilution 1:250, Invitrogen^TM^, Thermo Fisher Scientific) for 2 h at RT. Then, cells were stained with 6-diamino-2-phenylindole (DAPI; 1:1000) and mounted with Vectashield antifade mounting medium (Vector Laboratories, Burlingame, CA, USA). Samples were observed using a fluorescence inverted microscope (Eclipse Ti-S, Nikon Imaging Japan Inc.) by the NIS-Elements D program (Nikon Imaging Japan Inc., Tokyo, Japan).

#### 4.8.2. Gene Expression Analysis

To determine mRNA expression of neurospheres, total RNA was isolated after 3, 7, 10, and 14 days of differentiation, using FavorPrep Tissue Total RNA Mini Kit (Favorgen Biotech Corp., PingTung, Taiwan) followed the manufacturer’s instructions. The RNAs were then reverse-transcribed using oligo-dT primers for cDNA synthesis (biotech rabbit GmbH, Berlin, Germany). Neurospheres-specific gene expressions were evaluated using KAPA SYBR FAST qPCR Master Mix (Applied Biosystems, Carlsbad, CA, USA). Gene expression was examined with QuantStudio 5 real-time PCR system (QuantStudio 5, Applied Biosystems, Carlsbad, CA, USA). Melting curve analysis was also performed to the specificity of the specific primers (Appendix A). *β-actin* was used as a reference gene to normalize the target genes, and the expression fold change was calculated relative to control cells, MSCs, and positive control, SHSY-5Y human neuroblastoma (was purchased from ATCC, #CRL-2266). qPCR was performed in triplicate and statistical analysis was performed using the 2^−∆∆CT^ method

### 4.9. Experimental Animals

Adult male Sprague Dawley (SD) rats (weight, 250–300 g or 8-weeks-old) were used in this study. All rats in this study were obtained from Nomura Siam International Co., Ltd., Bangkok, Thailand, kept in standard conditions under a 12-h light/dark cycle, and provided with food and water. All animal protocols were approved by the Institutional Animal Care and Use Committee at Suranaree University of Technology (8/2018).

The timeline of the SCI model and transplantation is shown in Figure 17B. Fifty male SD rats were selected and randomly divided into five groups—10 rats/group: (1) sham group; the rats were operated on but had no SCI induction, (2) control group; the rats were operated on to induce SCI and no cells transplant were performed, (3) normal saline group; the rats were operated on to induce SCI and vehicle treatment, (4) MSCs group; the rats were operated on to induce SCI and received hWJ-MSCs transplantation, and (5) NSs group; the rats were operated on to induce SCI and received NSs transplantation.

### 4.10. Spinal Cord Injury Model

The rat compression SCI model was performed according to procedures previously described [52]. Rats were anesthetized using a mixture of 1–3% isoflurane (Aerrane isoflurane, Baxter Healthcare Corporation, Deerfield, Illinois, USA) and 500 mL/min oxygen and maintained throughout surgery. Under the sterile technique, the compression SCI model was created by forceps compression; calibrated forceps with a spacer (1.3 mm) were used to bilaterally compress to the dura of the spinal cord at thoracic level 9 (T9) for 15 s [26,53]. Rats in the sham group were operated on with no SCI induction. Post operation, the rats were injected with 5 mg/kg tramadol hydrochloride, 5 mg/kg carprofen, 5 mg/kg enrofloxacin, and 3.5 mL 0.9% normal saline subcutaneous injections. Manual bladder expression was performed 3 times/daily until voluntary urination was established in combination with rats that were injected with 25 mg/kg sulfamethoxazole trimethoprim subcutaneous injections.

### 4.11. Cell Transplantations

NSs on day 7 from the Isx9 group were digested by 0.25% trypsin/EDTA at 37 °C, 5 min. Then, NSs and MSCs were harvested, washed by PBS(-), and prepared on ice until transplantation. The total concentration of cells for each group was 1 × 10^5^ cells in 3 mL normal saline. Cells transplantation was randomly performed at 9 days post-injury. Rats were anesthetized as described above under a sterile technique, and the spinal dura matter at T9 was exposed. Stem cells were intramedullary injected into the spinal cord using Hamilton’s needles (Hamilton Bonaduz AG, Bonaduz, Switzerland) as described previously [54].

For the NSs and MSCs groups, 3 µL of stem cells (1 × 10^5^ cells) suspension were transplanted into the spinal cord approximately 1.5 mm deep from the dorsal surface of the dura mater. For the normal saline group, rats were injected with 0.9% normal saline in the same protocol. In the last group, the sham group, the rats did not receive any specific treatment; only the spinal dura matter was exposed. All rats received the same medication as before.

### 4.12. Behavioral Tests

The BBB behavioral test method was used to evaluate the locomotor recovery of the SCI animals. It was developed by Basso, Beattie, and Bresnahan [31]. This method is a reliable and sensitive locomotor rating scale. The BBB behavioral test provides investigators with the performance measurement of behavioral outcomes after SCI and treatment. Briefly, all rats were placed on a floor in a square enclosure and recorded for 5 min. Their hindlimb locomotor functions were scored using a 22-point (0–21) scale which evaluated parameters including hindlimb joint movement, paw placement, weight support, forelimb hindlimb coordination, etc. Eight rats were selected randomly from each group and were evaluated with the BBB behavioral test 1 day before and after the SCI, and every week until the experiment was completed. The scores were obtained by averaging both hindlimb values.

### 4.13. Histology

To evaluate the pathology after 8 weeks of cell transplantation in the injured spinal cords, all rats were euthanized by carbon dioxide fumigation. Then, rats were intracardially perfused with 0.9% normal saline followed by 4% paraformaldehyde (PFA) in 0.1 M PBS at pH 7.4. Spinal cord sections surrounding the T9 lesion site (approximately 1 cm long) were carefully removed from the vertebrae and fixed with 4% PFA overnight at 4 °C. After dehydration, tissue was embedded in paraffin. Then, five spinal cord tissues were selected randomly from each transplanted group for cross-section and another five spinal cord tissues for sagittal section. The tissue series of slices were sectioned with 5 µm thickness and mounted on glass slides.

#### 4.13.1. Immunohistochemistry Staining

To evaluate the pathology after cell transplantation, the sagittal-sectioned tissue slides were deparaffinized by xylene and rehydrated. Heat-Induced Epitope Retrieval (HIER) was performed in an antigen retrieval reagent-basic solution (10 mM Tris-base, 1 mM EDTA and 0.05% tween 20, pH 9.0) for 10 min. After that, the sections were treated with 3% H_2_0_2_, in PBS, for 10 min. Then, the tissues were incubated in a blocking buffer (2% BSA, 5% normal goat serum, and 0.2% Triton X-100 in 0.1 M PBS) for 1h at RT.

To evaluate the survival and differentiation of the cells transplanted, IF was performed. Primary antibodies were diluted in blocking buffer following β-tubulin 3 (dilution 1:100), Nuclei antibody Cy3 conjugate (HuNu; for human nuclei, dilution 1:100), microtubule-associated protein-2 (MAP2; for neuron cells, dilution 1:100); glial fibrillary acidic protein (GFAP; for astrocytes; 1:100), Nestin (dilution 1:100); choline acetyltransferase (ChAT; for the synthesis of acetylcholine (ACh) in cholinergic neurons, dilution 1:100); neurofilament light chain (NF-L; for neurofilament light chain, dilution 1:100); and oligodendrocyte transcription factor Anti-antibody (Olig2; for mature oligodendrocytes and myelinating Schwann cells, dilution 1:100). Tissues were incubated in the primary antibodies solution at 4 °C overnight. The next day, the slides were stained with secondary antibodies Alexa fluor^®^ 488 donkey anti-mouse IgG (dilution 1:1000; Invitrogen^TM^, Thermo Fisher Scientific), Alexa fluor^®^ 594 goat anti-rabbit IgG (dilution 1:250; Invitrogen^TM^, Thermo Fisher Scientific), and Alexa fluor^®^ 594 goat anti-goat IgG (dilution 1:250; Invitrogen^TM^, Thermo Fisher Scientific) for 1h at RT and counterstained the nuclei with DAPI (dilution 1:1000). Then the tissue slides were mounted with the Vectashield mounting medium. The tissue slides were observed using a fluorescence inverted microscope.

To identify BDNF expression, immunoperoxidase or Immunohistochemical (IHC) staining was performed. Five sets of triple adjacent slices near the midline of the spinal cord were selected and prepared for BDNF expression measurements. After that, slides incubated in blocking solution were stained from immunoreactivity and neuroregeneration with anti-BDNF antibody (dilution 1:100) diluted in a blocking buffer. Slides were incubated in the primary antibodies solution at 4 °C overnight. The next day, slices were incubated with peroxidase-conjugated secondary antibody (dilution 1:100; Abcam) for 1 h at RT. After washing, the peroxidase activity sites were visualized with a 3,3′-diaminobenzidine tetrachloride kit (DAB; Vector Laboratories) for 5 min. Then, slides were dehydrated and cleared in xylene. Finally, slides were mounted with a mounting medium (Bio Mount HM, Bio-Optica, Milano, Italy) covered with coverslips. Negative control tissue sections (from the MSCs group) were processed similarly but the primary antibody was omitted. Finally, immunohistochemistry tissues were observed under a fluorescence inverted microscope. BDNF expression measurements were performed using Image J software. The brown particles in the picture were regarded as positive BDNF expression.

#### 4.13.2. Cavitation Analysis

Cross-sectioned tissue slides with 5 µm thickness every 100 µm were selected by a total of five slices centered over the injured epicenter of the spinal cord from each sample obtained. Sections were deparaffinized and rehydrated. Then, the sections were incubated in 0.1% Luxol Fast Blue solution (0.1% LFB) at 60 °C overnight. The next day, the tissues were differentiated in a 0.05% lithium carbonate solution (Li_2_CO_3_) until the tissue color turned blue for 10 s followed by 70% ethyl alcohol for 30 s. After that, tissues were stained with Mayer’s hematoxylin (Bio-Optica) for 30 s. Then, 0.5% eosin Y (Bio-Optica) was stained for 3 min and then washed. Tissues were dehydrated and cleared in xylene. Tissues were mounted with a mounting medium (Bio Mount HM, Bio-Optica, Milano, Italy) covered with coverslips.

Five tissue slices from each sample, including the lesion center of the spinal cord, were evaluated for LFB-positive areas. The slides were observed under an inverted microscope. The integrated optical density (I.O.D.) of the myelin sheath and the number of neurons in the spinal anterior horn were obtained using Image J software. To measure the cavitation area of each section, necrotic tissue spaces, which were found in the spinal cord area, were identified as part of the injured tissue, as the cavity area (Area.cav). The area of whole spinal cord tissue was measured as the total spinal cord area (Area.total) and calculated as percentage area cavitation (%Area.cav), which was evaluated by the formula:%Area.cav = Area.cav/Area.total × 100%

The volume of the injured spinal cord was calculated by using the Cavalieri method [55]. This method was calculated from the area of each section multiplied by the distance between each section and then calculated to determine the percentage volume cavitation (%Vol.cav), which was evaluated by the formula:%Vol.cav = Vol.cav/Vol.total × 100%

#### 4.13.3. Statistical Analysis

Statistical analysis was performed using GraphPad version 5 (GraphPad Software, San Diego, CA, USA), and data were expressed as the mean ± S.D. The differences between values were determined using a one-way analysis of variance (ANOVA), followed by the Tukey–Kramer Honest Significant Difference (HSD) Post hoc test to compare differences between the two groups. A value of *p* < 0.05 was considered significant with different lower-case letters. The data were plotted by GraphPad Prism version 5.

## 5. Conclusions

hWJ-MSCs differentiated to NSs by neurogenesis-enhancing small molecules was studied. Induction media composed of 10 µM Isx9 could induce hWJ-MSCs to be NSs by the Wnt3A signaling pathway. These NSs were shown to be neural stem/progenitor cells by gene expression such as *Nestin*, *β-catenin*, *β-tubulin 3,* and *NeuroD1*. Seven days after induction in the Isx9 medium, NSs were selected to be used in the treatment of spinal cord injury in the rat model. After transplantation, those NSs could survive and differentiate into neural lineages with neurotransmitter activity. Neurotrophic factors and axonal regeneration were also discovered in the NSs transplanted group. However, the MSCs and NSs transplantation groups showed no differences in the area of cavitation and the volume cavitation of spinal cord in the injury area. Stem cell treatment from this work also improved the recovery movement in the SCI rat model by the BBB locomotor rating result. However, specific inhibitor markers should be further examined after NSs differentiation for more data support to the Wnt3A signaling pathway. Additionally, more frequent examination of the pathology and histology at different periods of the SCI model is suggested so that a more detailed understanding of the mechanism of the improvement can be concluded.

## Figures and Tables

**Figure 1 ijms-24-03846-f001:**
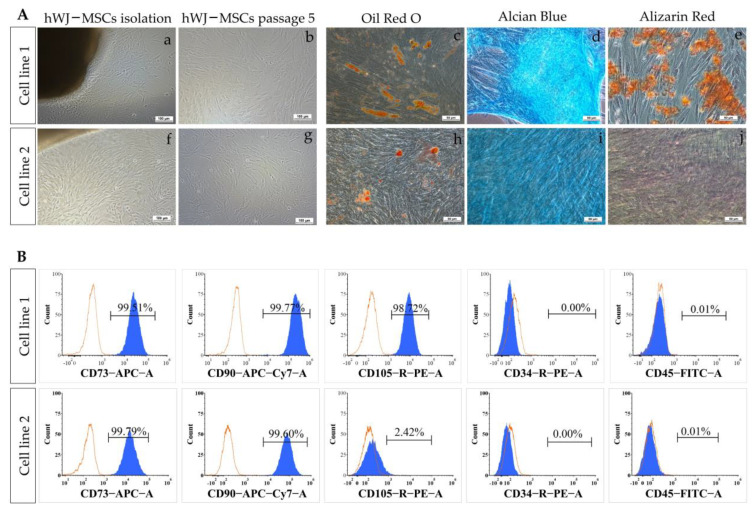
MSCs characterization. (**A**) Phase contrast images of MSCs expanded from Wharton’s Jelly tissue (a,f) and 80% confluences (b,g) of cell lines 1 and 2, scale bar = 100 µm. Trilineage differentiation ability of MSCs after 21 days (c–e, h–j), evaluated by Alizarin Red (osteogenic), Alcian Blue (chondrogenic), and Oil Red O (adipogenic) staining; scale bar = 50 µm. (**B**) Cell surface marker expression analysis of MSCs. MSCs of cell lines 1 and 2 were analyzed by flow cytometry with CD73^+^, CD90^+^, CD105^+^, CD34^-^, and CD45^-^ cell surface markers.

**Figure 2 ijms-24-03846-f002:**
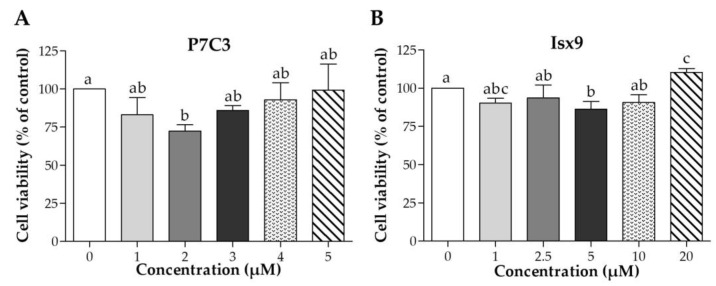
Cell viability of MSCs after P7C3 or Isx9 treatment. Cell viability was assessed by MTT assay at 0, 1, 2.5, 5, 10, and 20 µM for (**A**) P7C3 treatment for 48 h, and at 0, 0.25, 0.50, 1, 2.5, and 5 µM for (**B**) Isx9 treatment for 48 h. Data were shown as mean ± S.D. with different lower-case letters, and are significantly different at *p* < 0.05.

**Figure 3 ijms-24-03846-f003:**
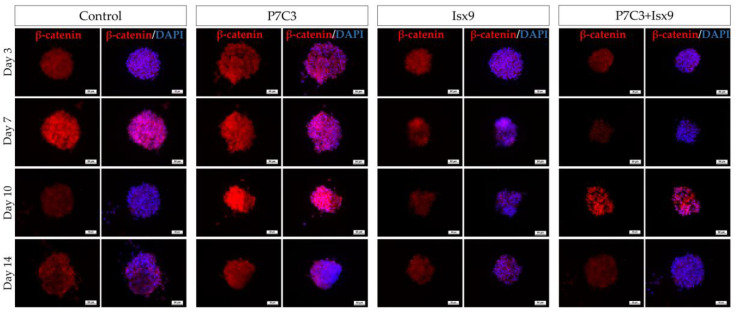
ICC staining image of NSs with β-catenin. NSs were co-stained with β-catenin (red; antigen-specific for Wnt3A signaling pathway activation) and DAPI (blue; nucleus), scale bar = 50 µm.

**Figure 4 ijms-24-03846-f004:**
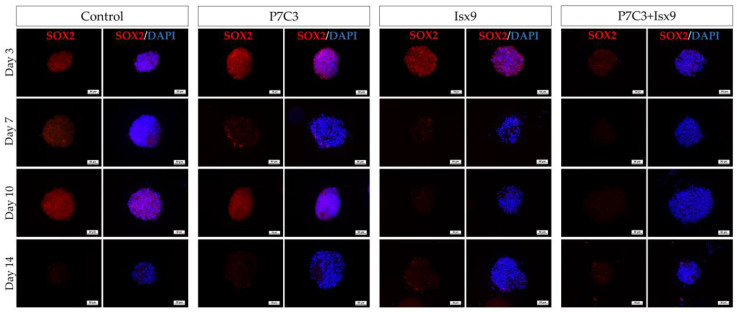
ICC staining image of NSs with SOX2. NSs were co-stained with SOX2 (red; antigen-specific for neural stem cell differentiation inhibitor) and DAPI (blue; nucleus), scale bar = 50 µm.

**Figure 5 ijms-24-03846-f005:**
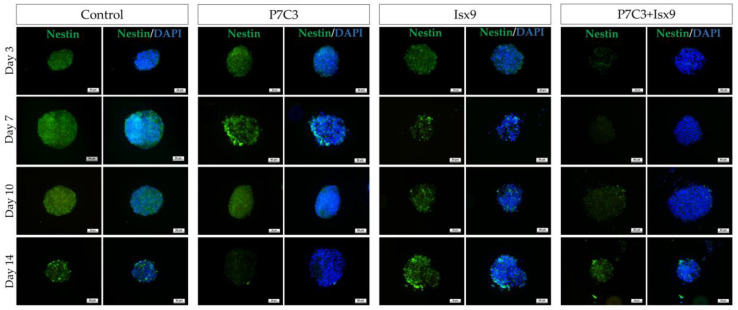
ICC staining image of NSs with Nestin. NSs were co-stained with Nestin (green; antigen-specific for neural stem/progenitor cells) and DAPI (blue; nucleus), scale bar = 50 µm.

**Figure 6 ijms-24-03846-f006:**
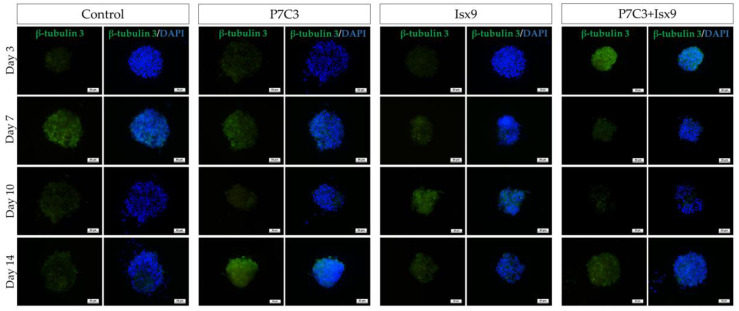
ICC staining image of NSs with β-tubulin 3. NSs were co-stained with β-tubulin 3 (green; antigen-specific for neuron cells) and DAPI (blue; nucleus), scale bar = 50 µm.

**Figure 7 ijms-24-03846-f007:**
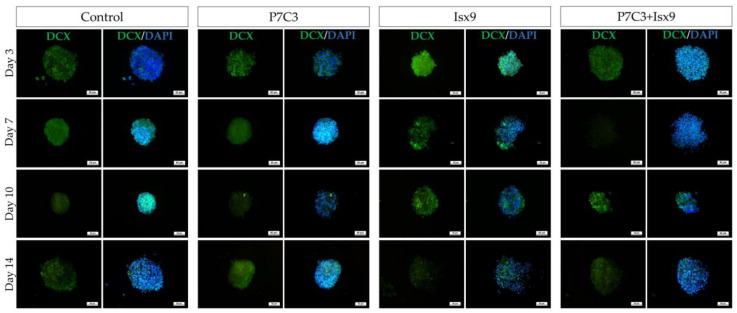
ICC staining image of NSs with DCX. NSs were co-stained with DCX (green; antigen-specific for immature neuron cells) and DAPI (blue; nucleus), scale bar = 50 µm.

**Figure 8 ijms-24-03846-f008:**
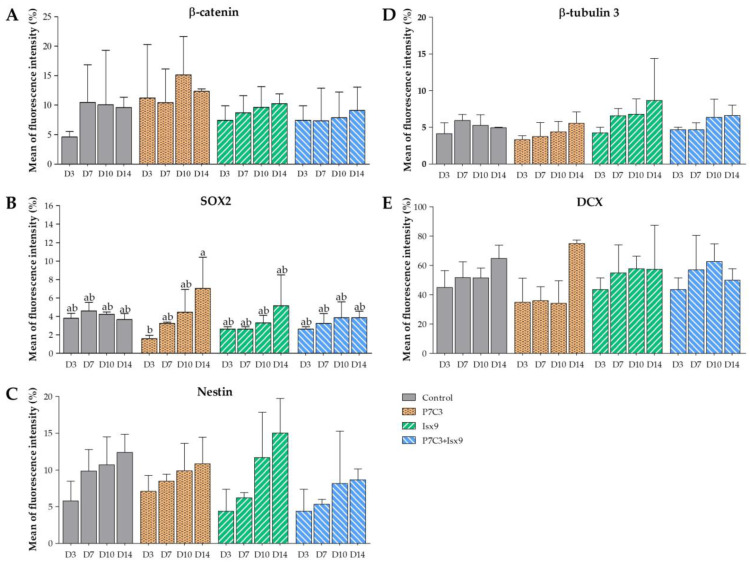
Immunofluorescence expression intensity results of NSs. (**A**) β-catenin, (**B**) SOX2, (**C**) Nestin, (**D**) β-tubulin 3, and DCX (**E**) intensity analysis were shown as mean ± S.D. with different lower-case letters, and are significantly different at *p* < 0.05.

**Figure 9 ijms-24-03846-f009:**
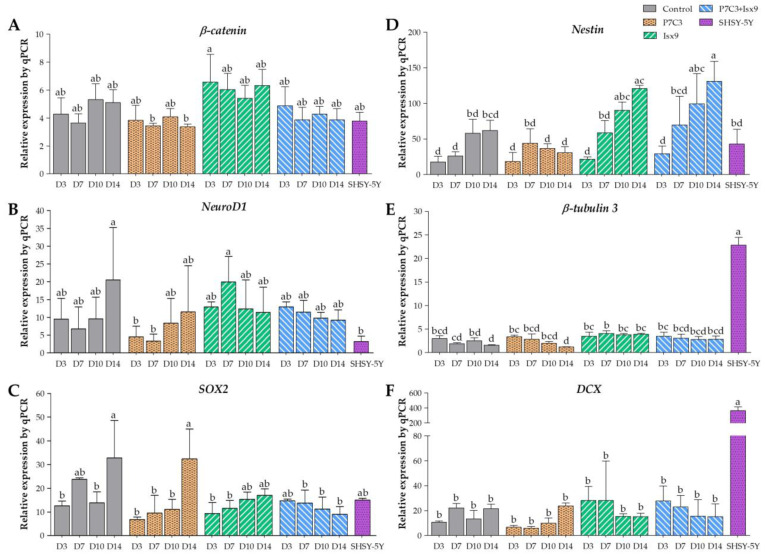
Gene expression analysis of NSs by qPCR, (**A**) *β-catenin*, (**B**) *NeuroD1* (**C**) *SOX2*, (**D**) *Nestin*, (**E**) *β-tubulin 3,* and (**F**) *DCX* genes. The target gene was normalized to *β-actin* as a reference gene and calculated the relative expression compared with each group. Data were shown as mean ± S.D. with different lower-case letters, and are significantly different at *p* < 0.05.

**Figure 10 ijms-24-03846-f010:**
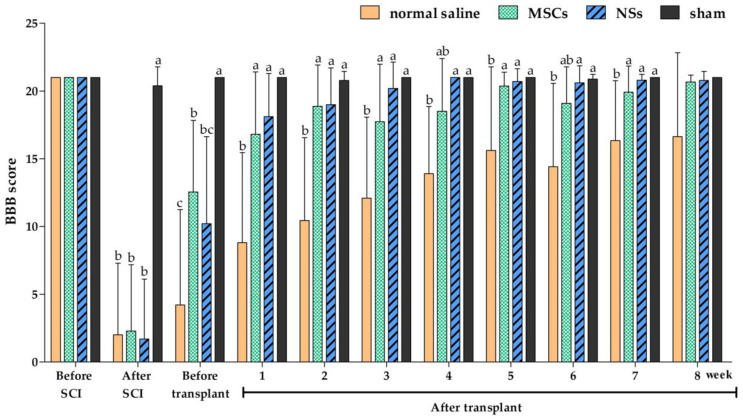
Behavioral test by the BBB locomotor rating score. The BBB locomotor rating scores were performed before and after the SCI, after stem cell transplantation, and weekly until the experiment was completed (n = 8 rats/group). Data were shown as mean ± S.D., with different lower-case letters, and are significantly different at *p* < 0.05.

**Figure 11 ijms-24-03846-f011:**
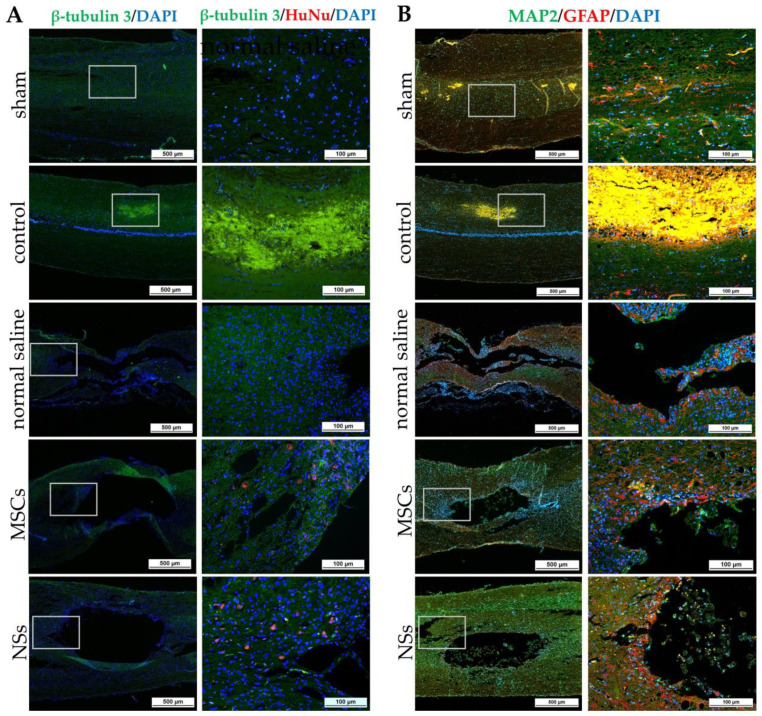
IF staining of SCI tissue 8 weeks after transplantation by β-tubulin 3, HuNu, MAP2, and GFAP. (**A**) Images of IF staining by β-tubulin 3 (green) co-stained with HuNu (red), scale bars = 500 µm (left) and 100 µm (right). (**B**) Images of IF staining by MAP2 (green) co-stained with GFAP (red), scale bars = 500 µm (left) and 100 µm (right).

**Figure 12 ijms-24-03846-f012:**
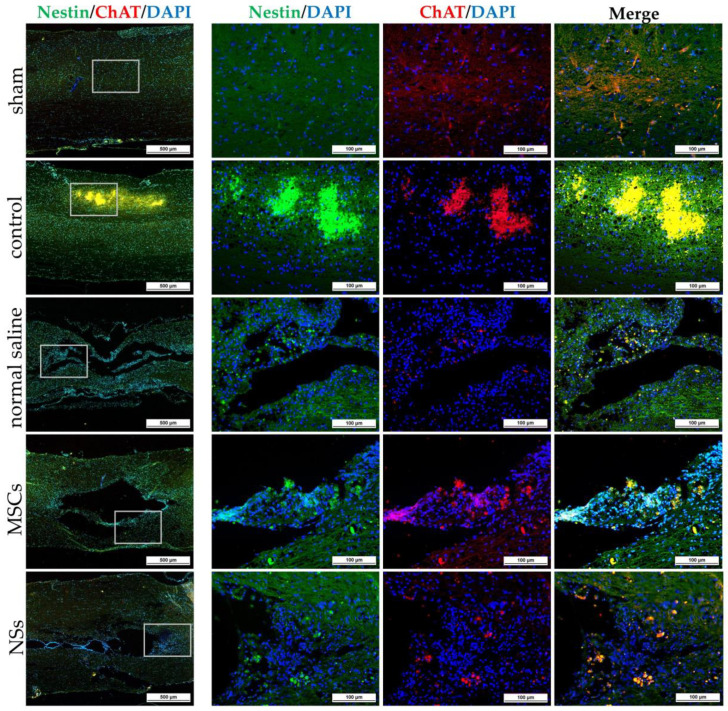
IF staining of SCI tissue 8 weeks after transplantation by Nestin (green) co-stained with ChAT (red), scale bars = 500 µm (left) and 100 µm (right).

**Figure 13 ijms-24-03846-f013:**
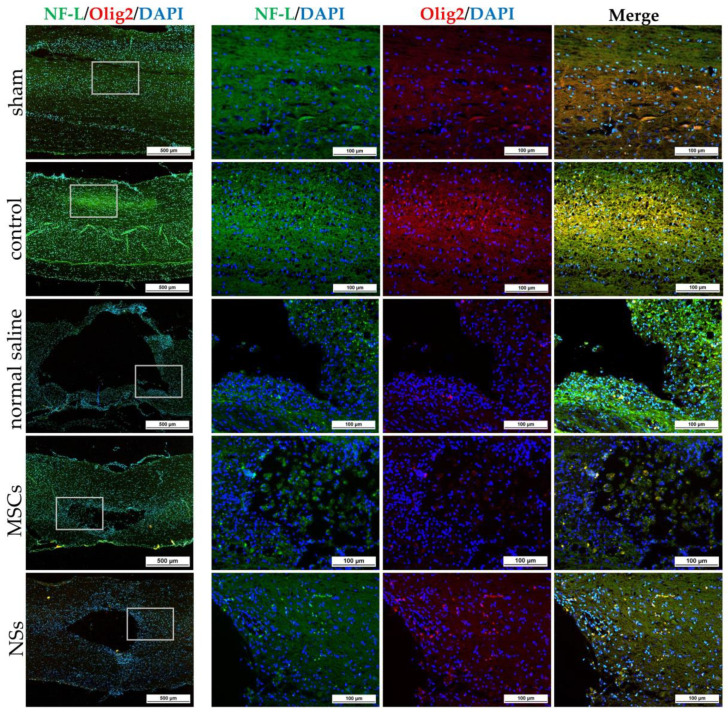
IF staining of SCI tissue 8 weeks after transplantation by NF-L (green) co-stained with Olig2 (red), scale bars = 500 µm (left) and 100 µm (right).

**Figure 14 ijms-24-03846-f014:**
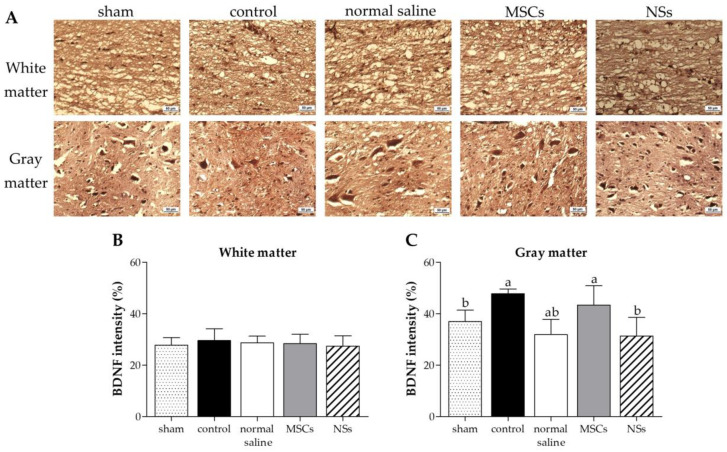
Immunohistochemistry (IHC) staining of SCI tissue at 8 weeks after transplantation. (**A**) IHC staining images of BDNF at white matter and gray matter; brown particles were converged around the cavity of the spinal cord, scale bar = 50 µm. BDNF intensity results at (**B**) white matter and (**C**) gray matter of SCI tissue. Data were shown as mean ± S.D. with different lower-case letters, and are significantly different at *p* < 0.05.

**Figure 15 ijms-24-03846-f015:**
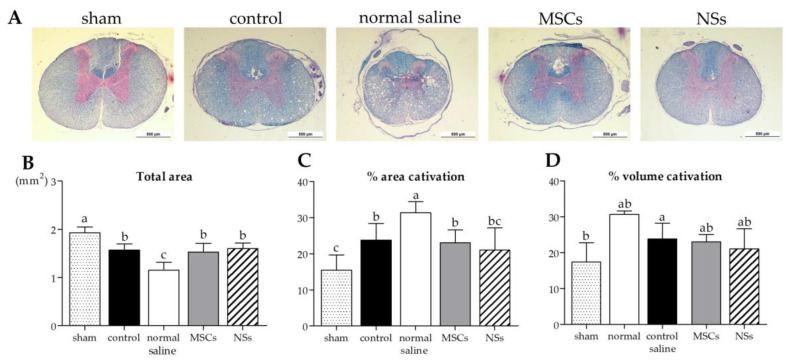
LFB/H&E staining of spinal cord tissue analysis. (**A**) Images of spinal cord tissue stained with LFB/H&E, scale bar = 500 µm. The results of cavitation analysis from LFB/H&E staining of spinal cord injury tissue, (**B**) total area of the spinal cord (mm^2^), (**C**) percentage of area cavitation, and (**D**) volume cavitation were shown. Data were shown as mean ± S.D. with different lower-case letters, and are significantly different at *p* < 0.05.

**Figure 16 ijms-24-03846-f016:**
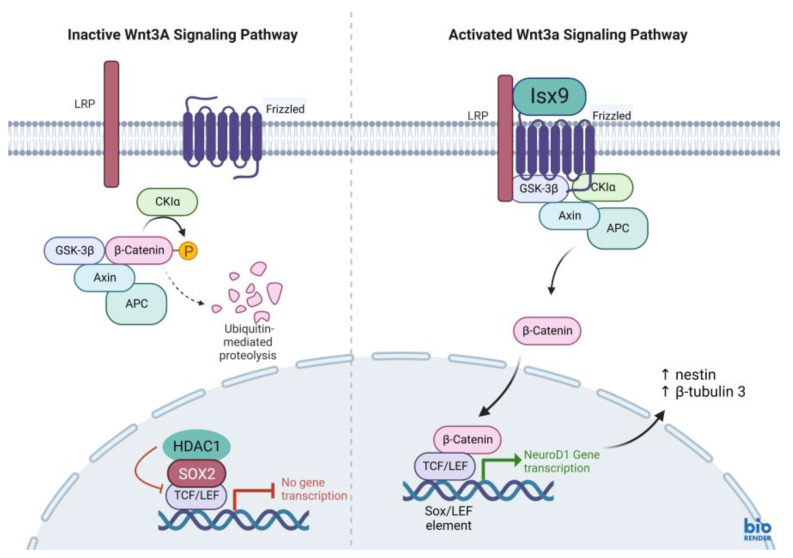
Wnt3A signaling pathway. (left) Inactive Wnt3A signaling pathway, β-catenin are degraded by multiprotein complex of ubiquitin-mediated proteolysis. (right) Activated Wnt3A signaling pathway. When the small molecule Isx9 binds to the LRP (lipoprotein receptor), then a multiprotein complex composed of Axin, APC, CK1, and GSK-3B is activated and β-catenin is released. β-catenin translocates through the nucleus and interacts with TCF/LEF transcription factor. While the repressor protein complex, HDAC1 and SOX2, is degraded. β-catenin and TCF/LEF regulate target gene transcription. *NeuroD1* gene transcription (neurogenic differentiation 1; NeuroD1), the target gene, and activated neurogenic differentiation markers such as Nestin and β-tubulin3. Created with BioRender.com.

**Figure 17 ijms-24-03846-f017:**
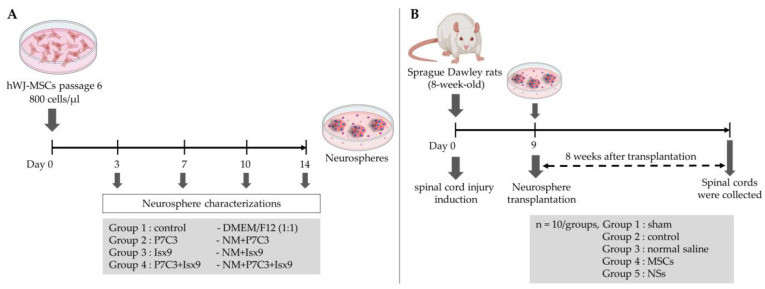
Experimental design of (**A**) neurospheres differentiation and (**B**) spinal cord injury transplantation timeline. Created with BioRender.com.

## Data Availability

Data included in this study are available upon request from the corresponding author.

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
