# Peer review of "The Efficiency of Neurospheres Derived from Human Wharton’s Jelly Mesenchymal Stem Cells for Spinal Cord Injury Regeneration in Rats"

_ijms, 2023, doi:10.3390/ijms24043846_

Round 1

Reviewer 1 Report

The Authors in this study aimed to induce hWJ-MSCs into neural stem/progenitor cells in sphere formation (neurospheres) by using neurogenesis-enhancing small molecules (P7C3 and Isx9) and transplant to recover Spinal cord injury (SCI) in rat model. Neurospheres induction were characterized by immunocytochemistry (ICC) and gene expression analysis. The Authors indicated that only neurospheres induced by 10 μM Isx9 for 7 days produced neural stem/progenitor cell markers such as nestin and β-tubulin 3 through Wnt3A signaling pathway regulation marker (β-catenin and NEUROD1) were also expressed. The neurospheres from the 7-day Isx9 group were selected to be transplanted into 9-day SCI rats. The Author indicated that, 8 weeks after transplantation, rats transplanted with the neurospheres could move normally. Moreover, MSCs and neurosphere cells were detected in the injured spinal cord tissue and produced neurotransmitter activity. Neurospheres transplanted rats showed the lowest cavity size of the SCI tissue resulted from injury recovery mechanism. The Authors concluded that hWJ-MSCs could differentiate into neurospheres using 10 μM Isx9 media through Wnt3A signaling pathway.

REVIEWER’S COMMENT

Although the work presented by the Authors considers an interesting topic in the field of spinal cord injury regeneration, the manuscript in its current form has major limitations of experimental setup, evaluation of results and conclusions.

Specifically, the major limitations and critical points include the following key aspects:

-Number of biological replicates of the cells

 It is note that Mesenchymal stem cells from Wharton's jelly of the umbilical cord can be obtained easily and proliferate rapidly in culture, and they are immunologically compatible and amenable to stable transfection. ( Chen, H., Zhang, Y., Yang, Z., & Zhang, H. (2013). Human umbilical cord Wharton's jelly-derived oligodendrocyte precursor-like cells for axon and myelin sheath regeneration. Neural regeneration research8(10), 890–899. https://doi.org/10.3969/j.issn.1673-5374.2013.10.003).

However, the Authors in this research use only 1 (one) population of hWJ-MSCs derived from umbilical cord of a single subject to form neurospheres. Rigor and scientific reproducibility of the data dictate that the results are the average of at least 3 biological replicates (3 cell populations derived from as many subjects/umbilical cords) exhibiting similar characteristics and behaviors to treatments.

 -Selection criteria for neurosphere cells for transplantation into SCI rat model

The Authors, with the great limitation of only one population used, choose to transplantat only neurosphere cells induced in the presence of 10 μM Isx9 for 7 days, based on the expression results of neural stem/progenitor cell markers such as nestin and β-tubulin 3, and also β-catenin, DCX and SOX2 expressions and their fluorescence intensity analysis (Figures 3-8). However, the values of these protein markers shown in Figures 8 are similar for all neurosphere groups (P7C3, Isx9 and P7C3+ Isx9) that at day 7 showed no statistically significant values. Moreover, also SOX2,protein and gene, presented similar low levels expression in all groups examined (Figures 8 and 9).

Based on these presented results, the Authors should have similarly considered the others neurosphere groups for transplantation and not just the neurosphere +Isx9 at 7 days. On the other and, they should have evaluated additional markers and/or more corroborating investigations to support their choice.

-Moreover, no data on NEUROD1 expression were showed as the Authors' statement in the abstract.

- Although the NSs+Isx9 cells efficacy in SCI recovery were evidenced in Figures 11-15, possible beneficial effects of NSs +P7C3 and/or  P7C3+ Isx9 cannot be ruled out after transplantation. This aspect is further important to draw reliable conclusions.

-The Authors also need to explain why in Figure 11 the MSC image (hWJ-MSCs) does not showed the expression of HuNu (marker for human nuclei). hWJ-MSCs transplanted cells should be present in these tissue samples..

- The Authors suggest a mechanism of action of the Wnt3a Signaling Pathway in the presence of Isx9 (Figure 16). But, this hypothesis needs to be testing. In order to demonstrate the activation of this mechanism in the presence of the molecule, treatments with specific inhibitors (e.g., LRP binging inhibitor or others) should be done. 

-Finally, the neurospheres immunomodulation (as Chemokines  an genes associated with immunomodulation and inflammation (Tnfα, IL-1β, Mrc1, Irf5 and Nf-κb), was not investigated/considered in the transplanted tissue or discussed in the manuscript.

Due to the considerations and limitations, lack of scientific method for the reproducibility of the data presented in the research, in the Reviewer's opinion, the Manuscript in its current form cannot be considered for publication.

A complete revision of the Manuscript is suggested.

Reviewer 2 Report

Dear authors.

The paper submitted for review deals with the important topic of the use of advanced therapy in spinal cord injury recovery. The use of stem cells in the treatment of spinal cord injury has been the subject of research at many centres and, despite many years of research, has not yielded positive results in the patient's clinical staging as opposed to olfactory glial cells.

The assessment of the work is excellent. I think it is one of the best prepared papers I have reviewed. The work was done very well from the outset, both in the preparation of the material, the execution of the study and the analysis of the results. The work also stands out in terms of editorial preparation, quality of illustrations. 

The results presented will make a significant scientific contribution to the development of regenerative medicine. 

Round 2

Reviewer 1 Report

This Reviewer understands the Authors' important technical difficulties working human cell samples. However, it is not scientifically sound to consider a single biological replicate in the experimental design. Lacking any assessment of biological variability this work is not scientifically robust.

I suggest trying to resubmit it in another form (Case Report?)
